# Issues Related to the Treatment of *H. pylori* Infection in People Living with HIV and Receiving Antiretrovirals

**DOI:** 10.3390/microorganisms10081541

**Published:** 2022-07-29

**Authors:** Marcel Nkuize, Stéphane De Wit

**Affiliations:** 1Department of Gastroenterology, CHU Saint Pierre, Université Libre de Bruxelles, 1050 Bruxelles, Belgium; 2Department of Infectious Diseases, CHU Saint Pierre, Université Libre de Bruxelles, 1050 Bruxelles, Belgium; stephane.dewit@stpierre-bru.be

**Keywords:** HIV, Helicobacter pylori, drug metabolism, interactions, coinfection, antiretroviral therapy, treatment issues, AISR concept, susceptibility testing, comedication, comorbidity

## Abstract

Treatment of *Helicobacter pylori* infection in people living with HIV is associated with several challenges, including those related to drug metabolism which plays a major role in treatment efficacy. In this review, we will discuss the enzymes involved in the metabolism of anti-*Helicobacter pylori* and anti-HIV drugs to provide a basis for understanding the potential for interactions between these drug classes. We will also provide a clinical perspective on other issues related to the treatment of *Helicobacter pylori* and HIV infections such as comorbidities, adherence, and peer communication. Finally, based on our understanding of the interplay between the above issues, we propose a new concept “Antimicrobial susceptibility testing-drug interaction-supports-referent physician” (AISR), to provide a framework for improving rates of *H. pylori* eradication in people living with HIV.

## 1. Introduction

*Helicobacter pylori (H. pylori)* infection affects more than half of the human population and more than 38 million people globally were infected with human immunodeficiency virus (HIV) as of 2021 [1,2]. The prevalence of *H. pylori* infection in people living with HIV (PLWH) varies by country with rates of 22% and 23% reported in China and the United Kingdom, respectively, 37% in Brazil, 47% in India, 70% in Iran, and 73% in Kenya [3,4,5,6,7]. *H. pylori* infection can cause severe complications such as gastric and duodenal ulcers (10–20%) and gastric cancer (1%) [1]. The frequency of these complications is influenced by various factors including immune characteristics. Duodenal and gastric ulcers appear to be less frequent in PLWH than in the general population, while gastric cancer has been reported to be 1.8-fold more frequent in PLWH than in the general population in South Korea, a region with a high risk of gastric cancer [3,8].

Successful *H. pylori* eradication can obviate these serious complications [1,9]. The treatment for *H. pylori* infection is well known and guidelines are available for the general population [1,10]. By contrast, despite the fact that co-infection has been observed for decades, there is a paucity of data regarding the treatment of *H. pylori* in PLWH. One study and one literature review on *H. pylori* treatment in coinfected people published recently by our team provide new data [11,12]. In our first large cohort study comparing response rates to *H. pylori* treatment in 258 PLWH and 204 controls, the overall eradication rates were 75.9% and 91.0%, respectively [11]. The subgroup treated with standard triple therapy against *H. pylori* presented with side effects that led the patient to modify the prescribed dose in 14/225 (6.2%) vs. 2/176 (1.1%) in coinfected PLWH and controls, respectively. HIV-positive status increased the risk of *H. pylori* treatment failure. The risk factors for *H. pylori* treatment failure in coinfected PLWH were the use of antiretrovirals, alcohol habits, and *H. pylori* treatment strategy [11].

While antiretroviral (ARV) therapies have dramatically increased the lifespan of PLWH, the prevalence of non-opportunistic infections, such as *H. pylori,* has been increasing over time with the increased aging of the HIV/AIDS population [13,14]. Treatment of both HIV infection and the patient’s additional comorbidities can result in polypharmacy and increase the frequency of drug interactions and adverse reactions [15,16,17]. Treatment of *H. pylori* infection in PLWH who are on ARV treatment, therefore, needs careful consideration. Indeed, drug metabolism is under high pressure in PLWH due to the fact that they are already undergoing lifelong treatment with a combination of ARVs that act at different stages of the HIV life cycle. In addition, PLWH also often receives treatment against opportunistic and non-opportunistic infections and for other co-morbidities such as cardiovascular disease, diabetes, and osteoporosis, for example [15,16].

In this setting, this review aims to, first, discuss issues related to the treatment of *H. pylori* infection in PLWH by discussing drug metabolism with an emphasis on the enzymes involved and the interactions between anti-*H. pylori* and antiretroviral drugs. Next, we will discuss, from a clinical perspective, the interplay between *H. pylori* and HIV infection treatments, comorbidities, treatment adherence, and peer communication. Finally, we will propose ways to tackle the issues related to *H. pylori* infection in PLWH in order to improve the efficacy of *H. pylori* infection treatment in these patients.

## 2. Methods

We performed a PUBMED and Google Scholar search using the keywords “drug or xenobiotic metabolism” AND “drug interactions or cytochrome P450” AND “antiretrovirals interaction” AND “ *Helicobacter pylori* or *H. pylori* treatment” AND “HIV co-infection” or “HIV comorbidities” AND “antiretroviral adherence” and “*H. pylori* treatment adherence or compliance” to identify literature published in English between January 2000 and March 2021. An additional manual search of internet sites on drug interactions was also performed.

## 3. Discussion

### 3.1. Metabolism of Anti-H. pylori Drugs

The treatment of *H. pylori* consists of a combination of an inhibitor of gastric acid secretion with two or more antibiotics and bismuth [1,10]. The metabolism of these drugs is discussed in this section.

#### 3.1.1. Metabolism of Gastric Acid Secretion Inhibitors

The metabolism of proton pump inhibitors (PPIs) involves cytochrome P450 (CYP) CYP3A4, CYP2C19, P-glycoprotein (P-gp), and a non-enzymatic mechanism (Table 1) [1]. For example, the PPI esomeprazole is a major substrate for CYP2C19, lansoprazole is a major substrate for CYP2C19, CYP3A4/5, and P-gp, and omeprazole and pantoprazole are major substrates for CYP2C19, and are minor substrates of CYP3A4/5. Rabeprazole is the sole PPI that is mainly metabolized by a non-enzymatic mechanism, but it is a minor substrate of CYP3A4/5 and CYP2C19. Equally important, there is a polymorphism of the gene for the CYP2C19 enzyme that leads to a mixed effect of PPIs on intragastric pH and *H. pylori* eradication across individuals [1,18,19,20].

A novel class of potent agents called potassium-competitive acid blockers block acid secretion by inhibiting the H^+^-K^+^-ATPase pump on parietal cells in the stomach. These drugs are metabolized by both oxidative (i.e., CYP) and non-oxidative pathways (i.e., sulfotransferase). Potassium-competitive acid blockers are more potent than PPIs [1,21].

#### 3.1.2. Metabolism of Anti-H. pylori Antibiotics

Antibiotics commonly prescribed against *H. pylori* infection include amoxicillin, clarithromycin, doxycycline, levofloxacin, metronidazole, rifabutin, and others [1,10].

Amoxicillin is a substrate of organic anion transporter 3 (OAT3) and is mainly eliminated unchanged by the kidney, whereas clarithromycin is a major substrate and major inhibitor of CYP3A4/5, and is a major inhibitor of P-gp (Table 1). Doxycycline is a minor inhibitor of CYP3A4/5, but its metabolism is not fully elucidated and 93% of the drug is excreted unchanged, whereas levofloxacin is a major inhibitor of CYP1A2, and a minor inhibitor of CYP3A4/5 and metronidazole is a major inhibitor of CYP2C9 [20]. Finally, Rifabutin (RFB) is a major inducer of P-gp and a moderate inducer of CYP3A4. However, intestinal CYP3A4 contributes significantly to RFB pre-systemic first-pass metabolism and drug interactions, notably with macrolide and antifungal agents [20,21,22,23,24,25,26]. The metabolism of bismuth is not well known [27].

### 3.2. Metabolism of Anti-HIV Drugs

The main classes of ARVs currently in use or in clinical development and their metabolism pathways are summarized in Table 1 [20,22,23,24,25].

Nucleoside reverse transcriptase inhibitors (NRTIs) are a component of most ARV combinations. NRTIs are all unlikely to cause either CYP inhibition or induction or to be metabolized by CYP (Table 1) [22]. Non-nucleoside reverse transcriptase inhibitors (NNRTIs) metabolically act as enzymatic inducers, inhibitors, substrates, or all of these (Table 1) [20,22,23]. Consequently, they affect the metabolism of many ARV and non-ARV drugs. Protease inhibitors act mostly as inhibitors of CYP (Table 1), impacting the metabolism of many co-administered drugs, which can be further increased by the addition of pharmacokinetic enhancers (PK enhancers) or boosters, such as ritonavir and cobicistat which are strong inhibitors of CYP3A, CYP2D6, and P-gp and may cause clinically significant alterations in serum levels of a variety of drugs metabolized by those pathways, including other antiretrovirals [20,22,23]. Integrase strand transfer inhibitors (INSTIs) are metabolized either by glucuronidation (dolutegravir, raltegravir), or by CYP3A4 inhibition (elvitegravir), or both glucuronidation and CYP3A4 (bictegravir and, to a lesser degree, dolutegravir), and by organic cation transporter 2 (dolutegravir) (Table 1) [15,22]. Cabotegravir is a novel oral long-acting INSTI, the substrate of the uridine glucuronyltransferase (UGT) enzyme [28]. Chemokine co-receptor antagonist, maraviroc, is a substrate of CYP3A and P-gp and has significant interactions with many medications (Table 1). Other classes include attachment inhibitors, such as fostemsavir, and post-attachment inhibitors, such as ibalizumab which is a monoclonal antibody that does not involve the CYP pathway (Table 1) [22,29,30].

Collectively, among ARVs, all PIs, all NNRTIs, and some INSTIs are mainly metabolized using the CYP-mediated pathway and most are either inhibitors or inducers that subsequently lead to pharmacodynamic or pharmacokinetic interactions with non-ARV substrates [22].

### 3.3. Drug–Drug and Drug–Food Interactions

A drug can be the substrate for enzymes that metabolize other substrates leading to the risk of drug–drug interactions (DDIs) [17,18,22,25,31,32]. Mechanisms of DDIs with CYP are mainly enzymatic induction (meaning increased synthesis or decreased degradation of the enzyme) and inhibition (which is either enzyme blockage or competition between drugs for the active site) [18,31,32]. Enzymatic induction results in decreased plasma concentration and pharmacologic effects of the drug. Enzymatic inhibition results in a reduction in the metabolism of the drug and prolongs its half-life, enhancing exposure to the drug or its metabolites and increasing the risk of side effects or toxicity [22]. However, this can be complex to predict as a drug can share inducer, inhibitor, and substrate properties with the same enzyme. In addition, if two drugs act similarly on the same enzyme, they may have additive pharmacodynamic effects if given concomitantly. By contrast, if two drugs have opposite actions on the same enzyme, they may antagonize the beneficial effects. Nonetheless, if several drugs are metabolized by the same cytochrome, there may be competition between substrates [17,18,22,31,32]. The effect of inhibitors is immediate after administration while the effect of inducers takes time to start and is delayed. In general, most of the ARVs metabolized via the CYP pathway are involved in DDIs. This may occur at the absorption, distribution, metabolism, or excretion level. DDIs can be avoided by forethought about possible impacts on treatment efficacy [17,18,22,31,32].

#### 3.3.1. Interactions of Anti-H. pylori Drugs and Antiretrovirals

Amoxicillin (AMX) and ARVs: AMX is a substrate of OAT3, is highly hydrophilic, and is mainly eliminated unchanged in the kidney. Based on metabolism and clearance, a clinically significant interaction is unlikely between AMX and all ARVs (Appendix A) [23,25,26].

Levofloxacin (LEV) and ARVs: LEV is a major inhibitor of CYP1A2 and a minor inhibitor of CYP3A4/5 [20]. Thus, significant pharmacokinetic DDIs between levofloxacin/ciprofloxacin and ARVs are unlikely (Appendix A), but there are two cautions. First, co-administration of quinolones (ciprofloxacin/LEV) with the NRTI lamivudine, which is eliminated by organic cation transporter 2 (OCT2), might potentially increase lamivudine concentrations due to inhibition of OCT2 by LEV [23].

Secondly, co-administration of quinolones should be avoided with simultaneous use of some PIs (atazanavir) or NNRTIs (rilpivirine, at supra-therapeutic dose) because of the risk of QT prolongation (Table 2) associated with each of them [17,20,22,23,33,34].

If coadministration cannot be avoided, an electrocardiogram before initiation and close monitoring is necessary. In addition, careful consideration should be taken to also rule out the concomitant use of drugs that carry a risk of QT interval prolongation such as antiarrhythmics, tricyclic antidepressants, and drugs for ionic imbalance. Finally, patients receiving either fluoroquinolones and/or zidovudine who have a medical history of glucose-6-phosphate dehydrogenase deficiency are at risk of hemolytic anemia, and this risk should be addressed [33].

Clarithromycin (CLA) and ARVs: CLA is a major substrate of CYP3A4/5 and a strong inhibitor of CYP3A4/5 and P-gp, leading to the potential for DDIs with numerous ARVs (Appendix A) [20,22,23].

Etravirine (CYP3A4 inducer) co-administration with CLA leads to decreased CLA effects and increased etravirine effects [20,23,25,36]. Maraviroc (substrate of CYP3A4 and P-gp) co-administration with CLA leads to increased maraviroc concentrations. In this case, and if creatinine clearance is below 30 mL/min or the patient is on hemodialysis, there is a requirement to adjust the maraviroc dose due to its partial elimination by the kidney [20,23,25,37]. Nevirapine (substrate and inducer of both CYP3A4 and CYP2B6) co-administration with CLA leads to a significant decrease in plasma concentration of CLA and decreases CLA efficacy [20,23,25,38]. Rilpivirine (substrate of CYP3A4) co-administration with CLA leads to increased rilpivirine concentrations, and supranormal doses of rilpivirine lead to a risk of QTc interval prolongation (Table 2), a risk which also exists with CLA alone [20,23,25,34]. Darunavir (substrate and weak intermediate inhibitor of CYP3A4)/cobicistat co-administration with CLA may result in increased CLA or darunavir/cobicistat plasma concentration, therefore an alternative treatment should be considered [20,23,25,39]. Efavirenz (substrate of CYP3A4, inhibitor of CYP2C19, CYP3A4, and P-gp, and inducer of CYP3A4) co-administration with CLA results in reduced plasma concentration and reduced effects of CLA [20,23,25,40]. Tenofovir alafenamide (TAF), the prodrug of tenofovir, is a substrate of P-gp, thus CLA, by inhibiting P-gp, is expected to increase the absorption of TAF and increase its systemic concentration. Similarly, tenofovir-disoproxil fumarate (TDF) (a substrate of P-gp) co-administration with CLA results in an increased systemic concentration of TDF. Monitoring of TDF and TAF reactions is recommended when co-administrated with CLA [20,23,25,41]. Finally, cobicistat (inhibitor of CYP3A, CYP2D6, P-gp) co-administered with CLA may increase plasma cobicistat concentrations and cobicistat may simultaneously increase serum levels of the co-administered CLA. CLA as a strong inhibitor of CYP3A4 and P-gp, should not be administered with PIs [20,23,25,39,42].

Metronidazole (MTZ) and ARVs: MTZ is a major inhibitor of CYP2C9 and co-administration with protease inhibitors (darunavir, lopinavir, ritonavir) leads to interactions that are linked to the alcohol included in the solution formulation, leading to the risk of a disulfiram-like reaction, which does not exist with tablets. A weak DDI, without clinical significance, has been found between MTZ and abacavir, and rilpivirine (Appendix A) [20,22,23].

Rifabutin (RFB) and ARVs: RFB is an inducer of CYP3A4 and is contraindicated with rilpivirine. DDIs are frequent with nearly all ARVs (Appendix A) with few exceptions. RFB may decrease cobicistat levels [20,25].

Proton pump inhibitors and ARV: PPIs, by increasing intragastric pH, may impair the absorption/solubility of protease inhibitors (PIs), particularly, atazanavir and NNRTIs, particularly rilpivirine (Appendix A), leading to a decrease in their systemic concentration [15,16,22,23,25]. Thus, for the treatment of *H. pylori* infection among PLWH receiving ARVs, drugs like atazanavir, and rilpvirine should be avoided because the use of half of the PPI dose, as suggested, will lead to suboptimal control of gastric acid secretion with subsequent failure to eradicate *H. pylori* infection [15,16,22,23,25].

#### 3.3.2. Drug–Food Interactions

Environmental factors can be inhibitors or inducers of CYP. Cabbage, broccoli, and polycyclic hydrocarbons found in tobacco and wood smoke are major inducers of CYP1A2 and may reduce the concentration levels and efficacy of drugs using this pathway, such as levofloxacin [20,33]. Nevertheless, broccoli has been suggested, inconsistently, to improve *H. pylori* eradication [43,44,45]. St John’s wort is an inducer of CYP3A, and grapefruit juice is an inhibitor of CYP3A found in the intestine [20,25]. These plants alter the metabolism of ARVs, but little is known about their effect on *H. pylori* drug metabolism.

Milk and cations contained in some drugs may significantly decrease the extent of absorption (by chelation) of tetracycline and levofloxacin [33,46]. PPIs are administered while fasting when intragastric pH is low, a condition known to trigger the prodrug to an active form of PPI at the level of the canaliculi of parietal cells [47]. Lifestyle factors could thus affect, via absorption effects, enzyme induction, and the efficacy of *H. pylori* eradication, but this has not yet been shown among PLWH.

### 3.4. Clinical Perspectives on H. pylori and HIV Therapies

From a clinical perspective, the efficacy of *H. pylori* eradication among PLWH may be influenced by the interplay of factors related to the bacteria, such as antimicrobial resistance, and the host, with particular attention to comorbidities, adherence to treatment, drug interactions, and inter-provider communication [1,11,15,16].

#### 3.4.1. *H. pylori* Infection Treatment: Antimicrobial Susceptibility Testing, DDIs, Adherence

Treatment of *H. pylori* infection can be challenging in PLWH. Coinfected PLWH carries *H. pylori* strains with higher rates of antibiotic resistance than the general population, which implies that management will depend on the availability of antimicrobial susceptibility testing (AST) [11,13].

If *H. pylori* AST is available, *H. pylori* treatment should also be guided according to the potential for DDIs (Appendix A) [11]. AST will also offer the possibility to choose rescue therapy in case of AST-guided therapy failure. If AST is unavailable, empirical anti-*H. pylori* treatment should include the most powerful anti-*H. pylori* drugs [12]. As drugs may be locally either not affordable or unavailable or contraindicated because of DDIs, a second-line empiric therapeutic choice will be based on the general clinical consensus for anti-*H. pylori* treatment in the general population as such a consensus does not exist for PLWH [1]. This second choice may, in turn, not be applicable due to a local high rate of *H. pylori* primary resistance to the antibiotic of choice.

In addition, other factors, such as the number of pills and frequency per day, and related side effects (e.g., abdominal pain with CLA, dysgeusia, disulfiram-like reaction with MTZ) and DDIs, vary depending on the drugs composing the *H. pylori* regimen, making the treatment simple or complex. These factors can affect treatment adherence and outcomes [1,11,48].

#### 3.4.2. HIV Infection Treatment: ARVs, Pill Burden, Side Effects

In the past, HIV treatment included a large number of pills that were taken on a daily basis [49,50]. There has been an improvement in ARVs with the advent of a single-pill combination of two to three ARV drugs that allow a reduction in the daily number of pills that must be taken (mostly available in high-income countries). Even so, currently, more than one pill daily is still necessary to achieve control of HIV viremia in low-income countries. Joint administration of ARVs with anti-*H. pylori* drugs could create a pill burden. The advent of long-acting drugs could be of great benefit to this issue [15,28,50].

Another factor is side effects. Overall, ARVs are well tolerated but side effects can be more frequent with a given ARV class such as abdominal pain, diarrhea with PIs, and weight gain with integrase inhibitors [15,16,17,22]. Additional side effects may be related to *H. pylori* drugs and the treatment of comorbidities as will be presented below [11,15]. The intensity of these combined side effects may be, in part, the consequence of DDIs, representing a threat that may impair compliance [11,15,16,17,19].

The use of a full dose of PPI is needed to obtain optimal control of acid secretion in the stomach that will potentiate antibiotic effects and treatment outcomes [1,47,51,52]. Thus, the treatment of *H. pylori* infection among PLHIV receiving ARVs including atazanavir should be avoided and an alternative ARV considered given [15,22,23,25]. In addition, careful consideration must also be given when macrolides and quinolones, and ARVs like PIs and rilpivirine are prescribed concomitantly (due to the risk of cardiac arrhythmia) [23,34,35]. These DDIs may affect the decision to prescribe one *H. pylori* treatment instead of another and could complicate therapeutic decision-making if options are limited in some regions [53,54].

#### 3.4.3. Comorbidities: Poly-Medication, DDIs, Adherence

Comorbidities are common in PLHIV receiving ARVs and concomitant treatment of those comorbidities may lead to multiple medications engaging in competition for CYP enzymes or transporters and DDIs [15,16,50]. In our experience, if the concomitant treatment for a given comorbidity is prescribed for a limited duration and DDIs are present, *H. pylori* treatment can be postponed until the concomitant medication ends or alternative ARVs can be transiently prescribed for the full duration of *H. pylori* treatment. Alternatively, if the concomitant medication is prescribed for an unlimited duration (e.g.,carbamazepine, diphantoin), one can consider an alternative *H. pylori* treatment or transiently change the ARV for the full course of *H. pylori* treatment.

Clearly, the need to take additional medications for comorbid conditions means that patients must frequently take a larger number of medications, increasing their risk of side effects and also potentially impacting treatment adherence, as is described below [15,16,49].

#### 3.4.4. Adherence

Adherence to medication as directed is one of the main determinants of the efficacy of treatment for *H. pylori* infection as well as for HIV [1,48,49,50]. With regard to adherence to ARV therapy, for example, a meta-analysis of 125 studies investigated the relative importance of different adherence barriers among 17,061 adults, 1099 children, and 856 adolescents and reported that forgetting medication was common and observed in 41.4% of adults, 63.2% of adolescents, and 29.2% of children/caregivers [49]. Other adherence barriers with ARV therapy included travel, toxicity, alcohol/substance misuse, and pill burden. Another meta-analysis of 19 randomized controlled trials [50], including 6312 adults with PLWH, investigated the impact of pill burden on adherence to ARV therapy and virological outcomes [50]. The meta-analysis concluded that a higher pill burden was associated with lower adherence (*p* = 0.04) and no viral suppression.

An early study by Graham et al. evaluated factors affecting *H. pylori* eradication in 93 individuals [48]. They showed that compliance with the prescribed medication is one of the main factors affecting *H. pylori* eradication. Indeed, those who took more than 60% of the prescribed medication eradicated *H. pylori* by 96% in contrast to 69% eradication in those who took less than 60%. A lack of compliance to antibiotic dosing, for example, can result in suboptimal plasma concentrations of the drug, leading to suboptimal efficacy, antibiotic resistance, and treatment failure with the eventual progression of gastritis to ulcer and carcinoma, and increased associated health care costs [1,9].

#### 3.4.5. Communication

In our practice, good communication between health professionals (i.e., gastroenterologists, infectious disease specialists, and pharmacists) who are involved in the daily care of PLWH is essential to optimize ARV and *H. pylori* treatment outcomes. Indeed, if DDIs occur between the proposed *H. pylori* treatment and ongoing ARVs, expert advice from an infectious disease specialist or pharmacologist may be necessary to propose alternative ARVs that are compatible with the planned *H. pylori* treatment, particularly for patients with prior ARV and anti-*H. pylori* drug issues (i.e., failures, side effects, drug resistance).

### 3.5. AST-Drug Interaction-Supports-Referent Physician (Acronym ‘AISR’) Concept

This concept (Table 3) has been developed to address the issues presented in this review (antibiotic resistance, drug interactions, pill number, side effects, and adherence) [11]. It was first described in 2021, based on a registry study of unmatched *H. pylori-*infected HIV-positive individuals (cases) and HIV-negative obese pre-bariatric surgery individuals (controls) [11]. Cases were enrolled from 2006 to 2017, controls from 2007 to 2014, and they received the standard of care against *H. pylori*. A supplementary “optimal” subgroup of 43 consecutive cases was enrolled prospectively from 2017 to 2019 and was treated based on antimicrobial susceptibility testing, drug interaction checking, and additional support (material and multimedia) by one physician. The eradication failure rate among the cases as a function of the therapeutic strategy was: “optimal” 4/42 (9.5%) vs. other (empirical and antibiogram-guided) 52/182 (28.5%). Among 225 cases treated with standard triple therapy, the multivariable analysis of the risk factors of *H. pylori* failure showed that the AISR strategy provides the best chance of eradication compared to other strategies.

AISR: The antimicrobial susceptibility testing (based on multiple sampling from the antrum and corpus) allows guided, individualized therapy and adapting the treatment according to DDIs allowing the use of the optimal dose of anti-*H. pylori* drugs (Figure 1), while support may be provided by the referent physician as described here.

The initial appointment aims to present, explain, and provide the objective of the treatment by using material support that presents the treatment schedule, duration, and dose of pills. The duration of undetectable HIV viremia may serve as the primary marker of adherence to the antiretroviral therapy and may help to evaluate the future adherence to *H. pylori* treatment. Furthermore, barriers to adherence to *H. pylori* treatment, as for HIV treatment, should be discussed, such as access to caregivers and psychosocial or financial issues [49,50]. In addition to making the patient aware of the relationship between treatment outcome and full adherence to dosing, frequency, and duration of treatment, information should be provided on the impact of lifestyles such as consumption of milk products that can lead to an impairment of quinolone and tetracycline metabolism, alcohol, and MTZ resulting in disulfiram reaction, and also inform them about common side effects and their treatment [23,33,46].

At treatment initiation for *H. pylori* infection (on day 2 or 3) multimedia support, such as phone/video calls and SMS messages, can be used to evaluate potential problems and provide solutions with regard to tolerance, side effects, and quality of life, to reassure, motivate, and, therefore, optimize adherence. A reminder by a family member or phone message can be a help when forgetting is a threat to medication adherence [49]. After treatment completion, the referent physician should explain in-depth and prescribe follow-up tests of two kinds: one is devoted to evaluating treatment outcomes; at this time, the new pieces of information to be provided should concern only prior specific conditions to consider (no antibiotics for 4 weeks, and PPI discontinuation for at least 2 weeks) to avoid false-negative results in follow-up testing (urea breath test, gastric biopsy, stools antigen). A false-negative result can be due to PPI inhibition of *H. pylori* urease while antibiotics slow the growth of *H. pylori* [1,51,52]. The second type of follow-up test to explain to the patient is upper gastrointestinal endoscopy for all gastric ulcers (after the treatment ends) and for precancerous lesions (after 2 to 3 years) [1,55].

This should allow the physician to propose treatment for failures and avoid ignoring persistent *H. pylori* infection or avoid missing control of precancerous lesions given the risk of progression to more severe disease over time in high-risk groups [1,9,55]. In our experience, the risk of missing follow-up testing is related to patients (i.e., travelers, those with a life-threatening condition either at the time of *H. pylori* infection diagnosis/treatment or after the *H. pylori* infection treatment completion, or those who are irregular at HIV ambulatory clinical visits, those who are followed by different physicians who were not involved in the request for upper gastrointestinal endoscopy, and those with chronic use of PPIs after the endoscopy), and physicians (not aware of the requested follow-up test). Thus, the role of the referent-physician in this task is central.

Overall, from our experience, the *H. pylori* eradication rate by the “AISR” concept is 38/42 (90.5%) to 38/39 (97.4%) by intention-to-treat and per-protocol analysis, respectively (Figure 1) [11]. Presumably, “AISR” minus “A” = “ISR” (i.e., if the antimicrobial susceptibly testing is unknown, the “ISR” concept may still be applicable since it tackles the other issues related to treatment (DDIs, supports to enhance the adherence, referent physician). These strategies are critical in low-income countries because second-line and salvage anti-*H. pylori* regimens may be unavailable and/or not affordable [53,54].

A limitation of our review concerns comorbidities since there is a wide spectrum of HIV-related and unrelated comorbidities that increases with aging and that cannot be exhaustively analyzed; there is also a paucity of literature on this topic to review. The strength of this paper is that, to the best of our knowledge, the current review is the first one describing issues related to, and a step-by-step approach to, *H. pylori* treatment in PLWH, and suggests a way to tackle issues related to this treatment, namely the AISR concept. This concept is drawn from the experience of one reference HIV/AIDS center that takes care of more than three thousand PLWH. AISR concept has become our standard of care for most of our patients and can be applied right now for use in clinical practice. We also point out some gaps in the current knowledge of this topic that necessitate further studies.

## 4. Conclusions

Among *H. pylori*-HIV co-infected patients who are on ARVs, the treatment of *H. pylori* infection is subject to issues related to *H. pylori* (antimicrobial resistance), the host (adherence to the treatment, genetics), drugs used for treatment (interactions, side effects), and the provider (communication). Furthermore, it is crucial to know the susceptibility of *H. pylori* strains as well as to rule out drug interactions before prescribing anti*-H. pylori* treatment because alternative treatment against *H. pylori* and HIV or dose adjustments may be required, and some anti-*H. pylori* and ARV combinations are contraindicated because of harmful side effects. Moreover, near-perfect adherence is also the cornerstone of anti-*H. pylori* treatment efficacy in *H. pylori*–HIV co-infected people. Hence, the ‘AISR’ concept (Antimicrobial susceptibility testing-interactions check-Supports-Referent-physician) is well-positioned to tackle those issues, and further large studies on the treatment of *H. pylori* among PLHIV receiving ARV from other regions are needed.

## Figures and Tables

**Figure 1 microorganisms-10-01541-f001:**
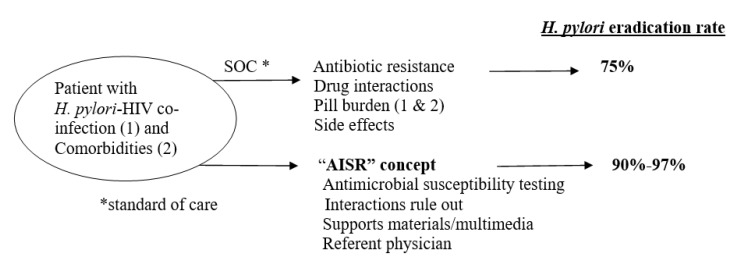
*H. pylori* treatment outcomes among *H. pylori*-HIV co-infected people receiving ARVs and treatment against comorbidities: *H. pylori* infection treatment as a function of strategies: standard of care versus “Antimicrobial susceptibility testing-Interactions check-Supports-Referent-physician”.

**Table 1 microorganisms-10-01541-t001:** Metabolic pathways of acid secretion inhibitors, antibiotics, and antiretrovirals (adapted from [20,21,22,23]).

Class, Subclass, Generic Name	Cytochrome (CYP)					Other Pathways
	3A4	2C19	1A2	P-gp	2C9	
**Acid secretion inhibitors**						
Esomeprazole	(s)	S, I				
Lansoprazole	S	S	(i)	S		
Omeprazole	(s)	S, I	(i)			
Pantoprazole	(s)	S			I	
Rabeprazole	(s)	(s)				Non-enzymatic
Vonoprazan	S	(s)				2 B6, 2D6, sulfotransferase 2A1
**Antibiotics**						
Amoxicillin						Organic anion transporter 3
Bismuth						Complex
Clarithromycin	S, I			I		
Doxycycline	(I)					Unknown
Levofloxacin	(I)		I			
Metronidazole					I	
Rifabutin	moi			I		
**Antiretrovirals**						
Nucleoside reverse transcriptase inhibitors (NRTIs)						
Abacavir						Cytosol ADH, uridine diphosphate glucuronyltransferase
Didanosine						Intracellular enzymes
Emtricitabine						Non-cytochrome pathway
Lamivudine						Passive diffusion or active uptake by transporters
Tenofovir alafenamide				S		
Tenofovir disoproxil F.					I	2E1 I
Zidovudine						Glucuronidation, passive diffusion, uptake transporters
Non-nucleoside reverse transcriptase inhibitors (NNRTIs)						
Efavirenz	S, I, i	I		I	I, i	2B6 S, I, i
Etravirine	S, i	S, I			S, I	
Delavirine	I					Other
Doravirine	S					
Nevirapine	I					
Rilpivirine	S			I		
Protease inhibitors						
Atazanavir	S, I		(I)	S, (I)		2C9(I),
Darunavir	S, I	(i)		I	(i)	
Lopinavir/Ritonavir	S, I/I			/I		/2D6 I, i other enzymes
Fosamprenavir	S,(I), i					
Nelfinavir	S, I	S, I	I		I	2B6, 2D6 I
Saquinavir	S, I			I		
Tipranavir	S, I, i4/5	I	I	S, I		2D6, organic anion transporter I
PK Enhancers (Boosters)						
Cobicistat	S, I			I		2D6 I
Ritonavir	S, I	I	I	I	I	2B6, 2C8, and 2D6 S: I
Fusion inhibitors						
Enfuvirtide						Proteolytic
Maraviroc	S			S		
Integrase strand transfer inhibitors (INSTIs)						
Bictegravir	S					
Dolutegravir	(s)					Organic cation transporter 2
Elvitegravir	S	(i)			(i)	
Raltegravir						Glucuronidation
Cabotegravir				S		Uridine diphosphate glucuronyltransferase
Attachment inhibitor						
Fostemsavir	(s)					Hydrolysis
Post-attachment inhibitor						
Ibalizumab						CD4-directed post-attachment antibodies

ADH, alcohol dehydrogenase, S substrate, (s) minor substrate, I inhibitor, (I) minor inhibitor, i inducer, moi moderate inducer, (i) minor inducer, OAT organic anion transporter, OATP organic anion transporter protein, OCT organic cation transporter.

**Table 2 microorganisms-10-01541-t002:** Major drug–drug interaction actors in *H. pylori* treatment among co-infected people living with HIV.

CYP inducers or inhibitors
ARVs: protease inhibitors, non-nucleotide reverse transcriptase inhibitors, integrase strand transfer inhibitors
Antimicrobials: macrolides, quinolones *, rifabutin
Major side effects for macrolides, and quinolones: QT prolongation, cardiac arrhythmia

CYP, Cytochrome P450; ARVs, antiretrovirals; * aortic dissection, rupture of the Achille’s tendon, phototoxicity [35].

**Table 3 microorganisms-10-01541-t003:** To be remembered when diagnosing and treating *H. pylori* infection in people living with HIV receiving antiretrovirals.

AISR Concept *
Antimicrobial susceptibility testing: This is the diagnostic test to obtain
Interaction search: Between anti-*H. pylori* drugs and antiretrovirals
Supports: Materials (handouts), multi-media (phone video call, SMS, cell phone reminder) to improve adherence
Referent physician: to manage follow-up tests (urea breath test, endoscopy for gastric ulcer and precancerous lesions)

* “ISR” if antimicrobial susceptibility testing is unavailable.

## Data Availability

Not applicable.

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
