# Peer review of "Issues Related to the Treatment of H. pylori Infection in People Living with HIV and Receiving Antiretrovirals"

_microorganisms, 2022, doi:10.3390/microorganisms10081541_

Round 1

Reviewer 1 Report

The papers tackles an interesting topic with great potential for improving patient care in the special context of H. pylori - HIV coninfection. However, to attain significant impact and scientific rigor, the paper needs to be majorly revised. 

- The introduction does not provide a convincing justification for the necessity of this review. What are the current statistics (if any) for H. pylori - HIV coinfection, response to eradication therapy, rate of complications compared to H. pylori alone? (example of a resource to obtain data https://onlinelibrary.wiley.com/doi/10.1111/hiv.13083 etc.)

- What does this review bring new compared to other reviews published so far (e.g.: https://www.microbiotajournal.com/article/673, https://pubmed.ncbi.nlm.nih.gov/24773336/) 

- Many informations lack references to support them, for example: "PPIs, by increasing intragastric pH, may impair the absorption/solubility of protease inhibitors (PIs), particularly, atazanavir and NNRTIs, particularly rilpivirine (Table 2), leading to a decrease in their systemic concentration."

- There are several sections with general theoretical data in the first half of the paper, not related to the H. pylori - HIV coinfection that miss the point of the review. There are plenty of resources about the pathophisiology of H. pylori alone or HIV alone infections. I suggest to significantly shorten the general, theoretical information data, and only point aspects relevant to the coinfection.

- Several informations lack specific data (exact statistical numbers) and only present general ideas not backed up by evidence-based data. For example: "Poor adherence remains a challenge globally both in low- and high-income regions, but its causes are variable". What studies say about some statistics regarding adherence? Is the adherence reffered to H. pylori infection treatment or HIV infection treatment.

- It is unclear when was the AISR concept first introduced? Is it based on a study? Which study? What was the study design, how many patients were included?

Author Response

Reviewer 1

We thank the reviewer for their comments that have improved the clarity of this manuscript. Below, the answer to each comment is provided.

Comments and answers (in red)

- The introduction does not provide a convincing justification for the necessity of this review. What are the current statistics (if any) for H. pylori - HIV coinfection, response to eradication therapy, rate of complications compared to H. pylori alone? (example of a resource to obtain data https://onlinelibrary.wiley.com/doi/10.1111/hiv.13083 etc.)

Thank you for this comment, we have added the following additional text to the introduction

Helicobacter pylori (H. pylori) infection affects more than half of the human population and more than 38 million people globally were infected with human immunodeficiency virus (HIV) as of 2021 [1,2]. The prevalence of H. pylori infection in people living with HIV (PLWH) varies by country with rates of 22% and 23% reported in China and the United Kingdom, respectively, 37% in Brazil, 47% in India, 70% in Iran, and 73% in Kenya [3,4,5,6,7]. H. pylori infection can cause severe complications such as gastric and duodenal ulcers (10%-20%) and gastric cancer (1%) [1]. The frequency of these complications is influenced by various factors including immune characteristics. Duodenal and gastric ulcers appear to be less frequent in PLWH than in the general population, while gastric cancer has been reported to be 1.8-fold more frequent in PLWH than in the general population in South Korea, a region with a high risk of gastric cancer [3,8].

Successful H. pylori eradication can obviate these serious complications [1,9]. The treatment for H. pylori infection is well known and guidelines are available for the general population [1,10]. By contrast, despite the fact that co-infection has been observed for decades, there is a paucity of data regarding the treatment of H. pylori in PLWH. One study and one literature review on H. pylori treatment in coinfected people published recently by our team provide new data [11,12]. In our first large cohort study comparing response rates to H. pylori treatment in 258 PLWH and 204 controls, the overall eradication rates were 75.9% and 91.0%, respectively [11]. The subgroup treated with standard triple therapy against H. pylori presented with side effects that led the patient to modify the prescribed dose in 14/225 (6.2%) vs 2/176 (1.1%) in coinfected PLWH and controls, respectively. HIV-positive status increased the risk of H. pylori treatment failure. The risk factors for H. pylori treatment failure in coinfected PLWH were the use of antiretrovirals, alcohol habits, and H. pylori treatment strategy [11].

While antiretroviral (ARV) therapies have dramatically increased the lifespan of PLWH, the prevalence of non-opportunistic infections, such as H. pylori, has been increasing over time with the aging of the HIV/AIDS population [13,14]. Treatment of both HIV infection and the patient’s additional comorbidities can result in polypharmacy and increase the frequency of drug interactions and adverse reactions [15-17]. Treatment of H. pylori infection in PLWH who are on ARV treatment, therefore, needs careful consideration. Indeed, drug metabolism is under high pressure in PLWH due to the fact that they are already undergoing lifelong treatment with a combination of ARVs that act at different stages of the HIV life cycle. In addition, PLWH also often receives treatment against opportunistic and non-opportunistic infections and for other co-morbidities such as cardiovascular disease, diabetes, and osteoporosis, for example [15,16].

In this setting, this review aims to, first, discuss issues related to the treatment of H. pylori infection in PLWH by discussing drug metabolism with an emphasis on the enzymes involved and the interactions between anti-H. pylori and antiretroviral drugs. Next, we will discuss, from a clinical perspective, the interplay between H. pylori and HIV infection treatments, comorbidities, treatment adherence, and peer communication. Finally, we will propose ways to tackle the issues related to H. pylori infection in PLWH in order to improve the efficacy of H. pylori infection treatment in these patients.

- What does this review bring new compared to other reviews published so far (e.g.:  https://www.microbiotajournal.com/article/673, https://pubmed.ncbi.nlm.nih.gov/24773336/) 

Answer: The two papers cited by the reviewer aimed first to suggest, on the basis of the current literature on HIV-negative people and on our trials in PLWH, the most suitable regimen against H. pylori for PLWH and emphasized the impact on the microbiota; the second one reviewed the prevalence and factors that may influence H. pylori infection in PLWH. To the best of our knowledge, the current review is the first one describing issues related to, and a step-by-step approach to, H. pylori treatment in PLWH, and suggests a way to tackle issues related to this treatment.

- Many informations lack references to support them, for example: "PPIs, by increasing intragastric pH, may impair the absorption/solubility of protease inhibitors (PIs), particularly, atazanavir and NNRTIs, particularly rilpivirine (Table 2), leading to a decrease in their systemic concentration."

Answer: The  references (in red) have been added in the appropriate places

-There are several sections with general theoretical data in the first half of the paper, not related to the H. pylori - HIV coinfection that miss the point of the review. There are plenty of resources about the pathophisiology of H. pylori alone or HIV alone infections. I suggest to significantly shorten the general, theoretical information data, and only point aspects relevant to the coinfection.

Answer: Thank you for this suggestion. This point has been addressed; 2 paragraphs “Drug metabolism” and  “Factors that modify metabolic activity” has been removed.

- Several informations lack specific data (exact statistical numbers) and only present general ideas not backed up by evidence-based data. For example: "Poor adherence remains a challenge globally both in low- and high-income regions, but its causes are variable". What studies say about some statistics regarding adherence? Is the adherence reffered to H. pylori infection treatment or HIV infection treatment.

Answer: Thank you for this suggestion. An additional text, in red, has now been added to Section 4.4 – Adherence.

- It is unclear when was the AISR concept first introduced? Is it based on a study? Which study? What was the study design, how many patients were included?

Answer:

Thank you for this suggestion. We have revised the text (in red) for the section that describes the AISR concept (Section 5).

Reviewer 2 Report

I recommend to accept the manuscript after minor revision.

There are only some points to correct:

 - please provide the list of abbreviations

 - please provide the number of ethical approval

  • - introduction and discussion section need improvement; please provide information on how your results will translate into clinical practice; 

- in discussion section please provide study strong points  and study limitation section

- please correct typos

All abovementioned issues are crucial for the credibility of the results. The paper can be accepted only after addressing all the issues and another subsequent review.

I recommend to accept the manuscript after minor revision.

Author Response

Reviewer 2

 We thank the reviewer for their comments, they improved the clarity of this manuscript. Below, answers to each comment are provided.

Comments and Answers (in red)

 - Please provide the list of abbreviations

Answer: The abbreviations are provided when the word appears for the first time in the review.

 - Please provide the number of ethical approval

Answer: This publication is a review that did not involve humans; therefore, ethical approval is not applicable.

-Introduction and discussion section need improvement; please provide information on how your results will translate into clinical practice

Answer: The introduction and discussion have been revised; please see the edits described above and shown in the manuscript in red. The AISR concept can be used now in clinical practice by applying each of A, S, I, and R.

- In the discussion section please provide study strong points  and study limitation section

Answer: This point has been addressed in the new version of the manuscript.

- Please correct typos

Answer: This point has been reviewed and improved.

Round 2

Reviewer 1 Report

Author's responded accurately to my suggestions. The article is publishable in the current form.

I would suggest the authors to also include in the manuscript this phrase: "To the best of our knowledge, the current review is the first one describing issues related to, and a step-by-step approach to, H. pylori treatment in PLWH, and suggests a way to tackle issues related to this treatment." to higlight the originality and increase impact.

Author Response

A limitation of our review concerns comorbidities since there is a wide spectrum of HIV-related and
unrelated comorbidities that increases with aging that cannot be exhaustively analyzed, and there is a paucity 
of literature on this topic to review. The strength of this paper is that, to the best of our knowledge, the 
current review is the first one describing issues related to, and a step-by-step approach to, H. pylori
treatment in PLWH, and suggests a way to tackle issues related to this treatment, namely the AISR concept. 
This concept is drawn from the experience of one reference HIV/AIDS center that takes care of more than 
three thousand PLWH. AISR concept has become our standard of care for most of our patients and can be 
applied right now for use in clinical practice. We also point out some gaps in the current knowledge of this 
topic that necessitate further studies.

Dear reviewer,

We thank you for the comments that have improved the strength of this manuscript. Below, the answer to each comment is provided.

Regards.

Comments and answers (in red)

I would suggest the authors to also include in the manuscript this phrase: "To the best of our knowledge, the current review is the first one describing issues related to, and a step-by-step approach to, H. pylori treatment in PLWH, and suggests a way to tackle issues related to this treatment."

Answer:

The strength of this paper is that, to the best of our knowledge, the current review is the first one describing issues related to, and a step-by-step approach to, H. pylori treatment in PLWH, and suggests a way to tackle issues related to this treatment, namely the AISR concept. This concept is drawn from the experience of one reference HIV/AIDS center that takes care of more than three thousand PLWH. AISR concept has become our standard of care for most of our patients and can be applied right now for use in clinical practice